# Infrared Evanescent Wave Sensing Based on a Ge_10_As_30_Se_40_Te_20_ Fiber for Alcohol Detection

**DOI:** 10.3390/s23104841

**Published:** 2023-05-17

**Authors:** Zijian Li, Yongkun Zhao, Tianxiang You, Jihong Zhu, Mengling Xia, Ping Lu, Xianghua Zhang, Yinsheng Xu

**Affiliations:** 1State Key Laboratory of Silicate Materials for Architectures, Wuhan University of Technology, Wuhan 430070, China; zijian_li@whut.edu.cn (Z.L.); 317330@whut.edu.cn (Y.Z.); ytxzs@whut.edu.cn (T.Y.); lupingwh@whut.edu.cn (P.L.); xzhang@univ-rennes1.fr (X.Z.); 2School of Materials Science and Engineering, Wuhan University of Technology, Wuhan 430070, China; zhujihong@yofc.com (J.Z.); xiamengling@hust.edu.cn (M.X.); 3State Key Laboratory of Optical Fiber and Cable Manufacture Technology, Yangtze Optical Fibre and Cable Joint Stock Limited Company (YOFC), Wuhan 430073, China; 4Institut Des Sciences Chimiques de Rennes UMR 6226, Centre National de la Recherche Scientifique (CNRS), Université de Rennes 1, 35042 Rennes, France

**Keywords:** fiber evanescent wave sensing, tapered fiber sensor, chalcogenide fiber, ethanol detection

## Abstract

Infrared evanescent wave sensing based on chalcogenide fiber is an emerging technology for qualitative and quantitative analysis of most organic compounds. Here, a tapered fiber sensor made from Ge_10_As_30_Se_40_Te_20_ glass fiber was reported. The fundamental modes and intensity of evanescent waves in fibers with different diameters were simulated with COMSOL. The 30 mm length tapered fiber sensors with different waist diameters, 110, 63, and 31 μm, were fabricated for ethanol detection. The sensor with a waist diameter of 31 μm has the highest sensitivity of 0.73 a.u./% and a limit of detection (LoD) of 0.195 vol.% for ethanol. Finally, this sensor has been used to analyze alcohols, including Chinese baijiu (Chinese distilled spirits), red wine, Shaoxing wine (Chinese rice wine), Rio cocktail, and Tsingtao beer. It is shown that the ethanol concentration is consistent with the nominal alcoholicity. Moreover, other components such as CO_2_ and maltose can be detected in Tsingtao beer, demonstrating the feasibility of its application in detecting food additives.

## 1. Introduction

Ethanol is one of the key substances widely used in the production process of medical, industrial, food, and other fields as an indispensable component or additive [1]. Ethanol is not only the most commonly used organic solvent, antidote, and anti-bacterial agent in industrial production but also a very effective preservative and disinfectant in daily life. In recent years, the development of environmental protection concepts has made it possible to find a renewable and clean energy alternative to gasoline. In addition, ethanol is also the main source of aroma in alcoholic beverages and an additive in many foods [2]. Therefore, it is important to detect alcoholicity in food accurately [3]. Traditionally, gas chromatography is a standard method used for liquid food testing, yet the complex sample preparation makes it, not real-time technology [4,5]. The emergence of a Fourier transform infrared spectrometer with attenuated total reflection (FTIR-ATR) truly reduces the complexity of sample preparation to a certain extent, but the sensitivity and accuracy of detection are greatly limited by ATR crystals [6,7,8,9]. Great demands for qualitative and quantitative analysis with low cost promote the development of fiber sensors, which are much more sensitive and flexible for liquid food testing [10].

Fiber sensors are extraordinary potential in various fields of water analysis, reaction kinetics analysis, food testing, etc. [11,12,13,14]. There has been some progress in the detection of ethanol concentrations in food with quartz optical fibers [15]. However, the limited transmission range of quartz fiber sensors is not suitable for analyzing multiplex organic compounds in liquid food [16,17,18]. Fortunately, chalcogenide fiber sensors based on evanescent waves can be an ideal detection method due to the advantages of a wide infrared transmission range (2–12 μm), high sensitivity, and high sample adaptability [19,20]. Moreover, the size can be reduced, and the sensitivity can be improved furtherly through geometry modifications because of the lower glass transition temperature (*T*_g_) of chalcogenide fiber (*T*_g_ < 300 °C) [21,22,23].

The evanescent wave fiber sensor works through the interaction between evanescent waves and external substances. The light wave propagating in the optical fiber follows Maxwell’s Equations. As shown in Figure 1, the light wave will not stop immediately at the interface between the fiber and external medium due to the continuity of the electromagnetic wave, and an electromagnetic wave called an evanescent wave would be generated on the side with low refractive index media [24]. After the action of external substances, the light will continue to transmit along the fiber with material information (molecular vibration/rotation information). The corresponding material information can be obtained for qualitative and quantitative research by analyzing output light. According to previous research, the penetration depth (*d*_p_) of the evanescent wave is an important parameter to describe the evanescent wave, and the expression can be described as [25],
(1)$$d_{p}=\frac{\lambda_{i}}{2\pi \sqrt{n_{1}^{2}\sin^{2}\theta_{i}-n_{2}^{2}}}$$
in which *λ*_i_ and *θ*_i_ are the wavelength and incident angle of the incident light, and *n*_1_ and *n*_2_ are the refractive index of the core and cladding, respectively. For bare fiber, *n*_2_ is the refractive index of the external environment. For fiber evanescent wave sensing (FEWS), the refractive index of fiber and sample remain unchanged; thus, the common method to increase the sensing performance of evanescent wave is to reduce *θ*_i_ by geometry modification, such as reducing the fiber diameter or bending the fiber [26].

Optical fiber sensors for ethanol analysis have appeared since the 1980s, while the study of quantitative analysis of ethanol solutions based on evanescent waves has developed rapidly in recent years [27,28]. Te_2_As_3_Se_5_ (TAS) fiber with a diameter of 400 μm was partly tapered into 200 μm and then detected 5 vol.% ethanol solution successfully, proving the feasibility of ethanol detection using tapered chalcogenide fiber [29]. Many improvements have been made in the preparation and process of tapered chalcogenide fiber, enabling the detection of ethanol, methanol, and various organic substances. However, the minimum detectable concentration of ethanol and the sensitivities of most sensors were not sufficient to meet the conditions for practical applications [30,31]. Wang et al. demonstrated that the smaller the waist diameter and the longer the taper waist length, the better the sensitivity. The sensitivity of the 15.94 mm taper waist length was more than 3 times as large as that of 11.53 mm [26]. Jiang et al. demonstrated that ethanol, tocopherol (vitamin E), ascorbic acid (vitamin C), fresh orange, and lemon juice could be detected using Ge–Te–Se glass fiber. However, the transmission band used for detection was just about 2–7 μm, ignoring much information within the molecular fingerprint area [32]. Recently, Vladimir shiryaev et al. prepared a fiber sensor with a replaceable probe of Ge_20_Se_80_ packaged in ceramic rods and ceramic discs, achieving the detection of ethanol and methanol while the theoretical limit of detection (LoD) can be 0.3% [33]. However, the sensitivity and LoD were not enough to support the quantitative detection of trace ethanol and illegal substitutes (such as methanol) in alcohols. The application of evanescent wave sensors for the detection of alcohols on sale is also in a state of scarcity [34].

In this work, we developed a tapered Ge_10_As_30_Se_40_Te_20_ (GAST) infrared fiber sensor with a waist diameter of 31 μm based on the results of COMSOL simulation. The sensitivity and LoD for ethanol detection can reach 0.73 a.u./% and 0.195 vol.%, respectively. The main composition and ethanol concentration of various alcohols, such as wine, beer, and hard liquors, were analyzed by the fiber sensor. In addition to ethanol, trace amounts of CO_2_ and maltose in Tsingtao beer can be detected as well, proving that this sensor can be used for food quality testing and chemical analysis.

## 2. Materials and Methods

### 2.1. Preparation of Ge_10_As_30_Se_40_Te_20_ Fiber

High-purity (5N) Ge, As, Se, and Te were used to prepare the Ge_10_As_30_Se_40_Te_20_ glass rod by melt quenching method [35]. Raw materials with 200 ppm Mg weighed in an Ar-protected glove box were firstly preheated to remove the absorbed water and volatile impurities and then placed in a distillation tube, which was slowly heated to 850 °C. All materials except Mg were distilled to the low hydroxyl silica ampoule during heating. Then the low hydroxyl ampoule tube was separated from the distillation tube by a flame torch. The distilled mixture in the sealed ampoule was then melted at 850 °C for 10 h and quenched in the air to form a glass rod [36]. After annealing at around 180 °C, a preform with a diameter of 14.7 mm and a length of 100 mm was obtained. A 1.5 mm thick glass disk cut from the preform was double-sided polished and tested using an FTIR spectrometer (INVENIO S, Bruker, Karlsruhe, Germany) to measure the optical transmission. Polyphenylene sulfone resin (PPSU) film (*T*_g_ = 220 °C) was evenly wrapped on the preform to avoid damage. Finally, a GAST/PPSU fiber with a diameter of 380/400 μm was obtained using a drawing tower. The fiber loss was obtained by the cut-off method [32,37].

### 2.2. Preparation and Encapsulation of Tapered Fiber

Tapered fibers were prepared by the fused taper method modified from the heat-brush approach [38]. The equipment for fiber tapering is shown in Figure 2a. Two linear translation stages with integrated stepper motors (NRT150/M, Thorlabs, Newton, NJ, USA) were used to draw the fiber. A ~10 cm length fiber was fixed by two fiber holders on these stages. A heating block in the middle of the fiber can raise the temperature to 360 °C was used to heat the fiber. A digital microscope was used to observe the taper drawing process of the fiber in real time, and an optical microscope was used to measure the parameters of tapered fiber more accurately after tapering completion.

The fiber is soaked in N,N-Dimethylformamide (DMF) solution for one hour to remove the PPSU protective layer before tapering. The temperature of the heating block is raised to a suitable temperature (350–360 °C) until the fiber deforms slightly, which is the time to start the taper process. The motion program of the moto steppers can be set in advance, and all parameters, including direction, speed, distance, and dwell time, can be controlled independently. As shown in Appendix A, the tapered fiber can be divided into three parts, untapered fiber, taper transition, and taper waist. The untapered fiber kept its original size, which was beneficial to coupling. The taper transition connecting the untapered fiber and the waist region had a length of 2 mm on each end. To keep the sensing region the same, the untapered fiber and the transition region were encapsulated in the UV-curing adhesive. As shown in Figure 2b, tapered fibers with unchanged waist length (*l*_w_ = 30 mm), whose waist diameters (*d*_w_) were 110, 63, and 31 μm, respectively, were obtained. The tapered fibers were directly encapsulated in the customized liquid pool without removing from the platform to keep the fiber straight. Encapsulated sensors with end faces cut flat were prepared, as shown in Figure 2c. The fiber with smaller *d*_w_ can be obtained, but it was too fragile to be tested.

### 2.3. Construction of Sensing Device

As shown in Figure 3, the sensing was carried out based on the FTIR spectrometer and mercury cadmium telluride (MCT) detection. Since the diameter of the beam from the spectrometer was too large to be coupled into the fiber efficiently, a plane-reflecting gold mirror and an off-axis parabolic gold mirror were used for reflection and first focusing. The beam was then further focused by a ZnSe lens with an extremely short focal length (6 mm) to make the beam available for coupling. It can be seen that both the sensor and detection were fixed to the three-dimensional translation stage, which can be adjusted freely to ensure the sensing efficiency. Each end face of the tapered fiber was cut flat to reduce coupling loss. Moreover, the MCT detector and the translation stage at the rear end are fixed on the same additional breadboard so that the length of the sensing fiber can be unlimited and suitable for fiber loss tests.

When the light transmits through the taper area full of liquid, the evanescent wave interacts with the external liquid, carrying material information, and then transmits along the fiber continuously. The outgoing signal is received by the MCT detector and analyzed by the computer to obtain the evanescent wave sensing spectrum. The spectrum transmitted by the fiber without any liquid is taken as the background spectrum, and different concentrations of liquid to be measured were added into the liquid pool from low to high. Some positions in the evanescent wave absorption spectra have obvious absorption peaks due to the absorption of evanescent waves with substances in the external environment, which can be used for qualitative analysis of substances. The changes in peak intensity (or peak area) due to the changes in concentrations of samples can be used to figure out the relationship between the concentration of liquid and the absorbance so as to calculate the sensitivity of the fiber sensor.

### 2.4. Preparation of Liquid Samples

Ethanol and ultrapure water were added to volumetric flasks in proportion using a pipette gun to prepare ethanol solution of different concentrations. The concentrations of prepared ethanol solution were 1, 10, 20, 30, 40, 50, 60, 70, 80, 90, and 99.5 vol.%, respectively. The concentration of ethanol is 99.5%.

Seven different alcohols, including Chinese baijiu A (54%, Luzhou Laojiao), Chinese baijiu B (42%, Mao Pu), Chinese baijiu C (40%, Joyboy), red wine (12.5%, Tini), Shaoxing wine (10.5%, Freshippo), Tsingtao beer (3.1%, Tsingtao Brewery), and Rio cocktail (3%, Rio) were detected with the encapsulated fiber sensor to determine their concentrations of ethanol.

## 3. Results and Discussion

### 3.1. Simulation of Fibers with Different Diameters

As the diameter of the fiber decreases, the evanescent wave increases. The distributions of modes and intensity of evanescent waves in fibers with different diameters were simulated using COMSOL, as shown in Figure 4. The mode set in the simulation was divided into two parts of fiber core with a high refractive index (*n*_1_ = 3.1) and the external environment medium with a low refractive index (*n*_2_ = 1.33). The wavelength of the incident light was set to 9.52 μm because this is the absorption peak of the C–O characteristic vibration in ethanol. The absorption and other losses of optical fibers have not been considered because the distribution and intensity of evanescent waves were explored by utilizing the light field in the cross-section of the fiber.

The distributions of fundamental modes in different fibers with diameters of 100, 50, and 30 μm are shown in Figure 4a–c. It is obvious that the pattern distribution gradually extends outward the fibers as the fiber diameter decreases, which explains the increase in evanescent wave. In order to study the intensity of the evanescent wave more intuitively, the variation of light intensity at the axis of the fiber cross-section is studied, and the intensity at the interface between the fiber core and the external environment is used as the intensity of the evanescent wave for analysis. The variation of light intensity is shown in Figure 4d,e, proving the increase in evanescent wave when fiber diameter decreases. It can be seen from the results that the reduction of fiber diameter can effectively enhance the intensity of evanescent waves to improve the sensing performance of fibers. This is because the total reflection angle of light decreases with the reduction of the fiber diameter, thereby enhancing *d*_p_ according to Equation (1). At the same time, the evanescent wave was influenced by high-order modes in the multimode fiber [24,31].

### 3.2. Properties of GAST Glass and Fiber

The infrared transmission spectrum of GAST glass is shown in Figure 5a. The transmission range of the glass covered 2–16 μm and can reach up to 57% at about 11 μm. There are almost no impurity absorption peaks in the glass, except for water (at 6.31 μm) and CO_2_ (at 4.24 and 4.27 μm), which proves the high purity of this GAST glass. The loss spectrum of GAST optical fiber is shown in Figure 5b. Impurity peaks of some sulfur oxides and hydrides appear in the spectrum, such as O-H (at 2.92 μm), Se-H (at 3.53 and 4.57 μm), As-H (at 5.02 μm), H_2_O (at 6.31 μm), Ge-O (at 7.9 μm), and As-O (at 8.9 μm) [20,30]. This is due to the inevitable oxidation during the purification, firing, and drawing processes. The fiber can transmit light effectively within the range of 2–10 μm, and the average loss can reach as low as 2 dB/m, which is within an acceptable range for the sensing experiment.

The end face cut flat was tested by scanning electron microscope-energy dispersive spectroscopy (SEM-EDS) to measure the diameter and component deviation of the fiber. As shown in Figure 6a, the diameter of the fiber without the PPSU layer is 376.6 μm. The slight undulating marks below were traces left by fiber cutting, covering a very small area that hardly affected the component testing and light coupling. Obviously, Ge, As, Se, and Te distribute homogeneously in the optical fiber according to the results of EDS mapping in Figure 6b. According to Table 1, the measured concentrations of these elements are consistent with those designed in advance.

### 3.3. Sensing Performance of Fibers with Different d_w_

Fiber tapering can effectively enhance the *d*_p_ of evanescent waves for the decrease in *θ*_i_ [31]. Three tapered fiber sensors with different *d*_w_, 110, 63, and 31 μm, have been prepared in this work. The sensing results of ethanol solution with different concentrations are shown in Figure 7. The intrinsic C–O stretching vibration bands of ethanol at 1047 and 1089 cm^−1^ are monitored to qualify the sensing performance because these peaks have high intensity and are not affected by the absorption peaks of water. It can be seen in Figure 7a–c that the absorption intensity of the same ethanol concentration increased with the decrease in *d*_w,_ and the peak at 1088 cm^−1^ weakened gradually until it could not be detected at the ethanol concentration of 1 vol.%. It can be found that the absorption peak area and the ethanol solution concentration obey the Lambert–Beer theory. The relationship between peak area and concentration was linear, and the slope of the line represented the sensitivity of the fiber sensor, as shown in Figure 7d. These areas are linearly dependent on the ethanol concentration, and the slope of the fitting line can be defined as the sensitivity of the sensor. The sensitivities of fibers with *d*_w_ = 110 and 63 μm are 0.35 and 0.58 a.u./%, respectively. As *d*_w_ decreases, the sensitivities of the tapered fiber sensors become higher, which can reach 0.73 a.u./% when *d*_w_ = 31 μm.

The LoD of fiber sensors can be calculated as,
(2)$$C_{\min}=3\sigma /S$$
in which *C*_min_ is the minimum concentration of detection, *σ* is the standard deviation of the peak intensity of the water tested 10 times, and *S* is the sensitivity of the fiber. The LoD of fiber sensors with *d*_w_ = 110, 63, and 31 μm can be 0.569, 0.358, and 0.195 vol./%, respectively. The improvements in the sensitivities and LoD can be attributed to the enhancement of the *d*_p_ of an evanescent wave for the decrease in *θ*_i_ in the tapered fiber, making the changes of evanescent wave absorption spectra more obvious, as shown in Figure 7e.

### 3.4. Detection of Ethanol in Alcohols

The excellent sensitivity of the sensor with *d*_w_ = 31 μm makes it be further applied in the ethanol detection of alcohols, including Chinese baijiu, red wine, Shaoxing wine, Tsingtao beer, and Rio cocktail. As shown in Figure 8, the evanescent wave absorption spectra of seven alcohols were obtained by the fiber sensor, of which the ethanol concentrations were 54, 42, 40, 12.5, 10.5, 3.1, and 3 vol.%, respectively. The baselines in spectra of different samples are offset due to the variation in the refractive index of different compositions of alcohols. Absorption peaks of ethanol and water can be clearly distinguished in Figure 8a, which shows a suitable regularity in the variation of ethanol concentrations in different alcohols. Peaks at 2340 cm^−1^ belonging to CO_2_ (used as an additive) can be detected in beer and cocktails.

More information in the “fingerprint area” from 1000 to 1200 cm^−1^ is shown in Figure 8b. For Chinese baijiu with high concentrations of ethanol, the detection of ν(C–O) can be clear and accurate, while it is much more difficult for the sensor to separate out ν(C–O) of ethanol from other vibration peaks of organic substances such as maltose in the beer. A noticeable widening in peaks of the cocktail and beer. Compared with the sensitivity curve calculated fitted before, it can be seen that the detected concentrations of these alcohols are very close to the original concentrations, as shown in Figure 8c. The detected concentrations of ethanol and calibrated concentrations in the ingredient lists are summarized in Table 2. All the measured ethanol concentrations are consistent with the nominal alcoholicity of the alcohols. The minimum deviations in detection can reach 0.1 vol.% for Red wine and Rio cocktail, showing the extremely high accuracy of the sensor. Interestingly, several small absorption peaks at 1000, 1022, 1119, and 1157 cm^−1^ can be found, as shown in Figure 8d, due to the maltose produced in the beer brewing process [39,40].

## 4. Conclusions

A tapered fiber evanescent wave sensor using Ge_10_As_30_Se_40_Te_20_ fiber was fabricated and applied to ethanol detection. The Ge_10_As_30_Se_40_Te_20_ fiber has the lowest loss at 1.8 dB/m at 5.6 μm and lower than 3 dB/m from 8.5 to 9.5 μm. The simulation results demonstrate that as the fiber diameter decreases, the fundamental mode of the fiber tends to diffuse outward, and the strength of evanescent waves increases. Fibers were then tapered into 110, 63, and 31 μm by fused taper method and packaged in designed liquid pools to obtain final sensors. Different concentrations of ethanol can be detected quantitatively. The sensitivities of these sensors were tested and calculated to be 0.35, 0.58, and 0.73 a.u./%, respectively. The LoD of fiber with *d*_w_ = 31 μm can reach 0.195 vol.%. The alcohols, such as Chinese baijiu, red wine, Shaoxing wine, Tsingtao Beer, and Rio cocktail, were analyzed by the sensor. The results showed that the ethanol concentration of these alcohols differed slightly from their nominal alcoholicity. The FEWS of Tsingtao beer and Rio cocktail revealed the presence of CO_2_ as a food additive. Maltose in the beer was also detected, showing the ability of this fiber sensor to analyze the trace substance in the food and other organic compounds.

## Figures and Tables

**Figure 1 sensors-23-04841-f001:**
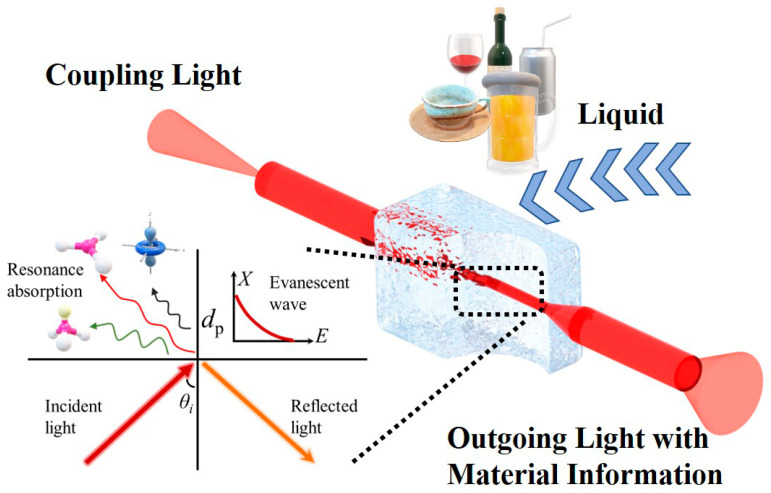
Principle of infrared fiber evanescent wave sensing. The total reflection of light occurs at the fiber interface, and the generated evanescent wave resonates with the chemical bond of the external liquid to output the evanescent wave absorption spectrum, which can be analyzed.

**Figure 2 sensors-23-04841-f002:**
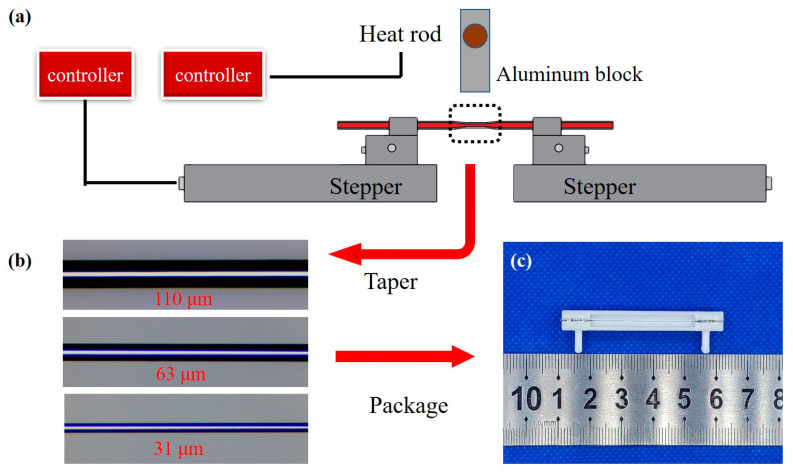
Diagram of sensor preparation. (**a**) Tapered platform with steppers and heat block. (**b**) Tapered fiber with different *d*_w_ = 110, 63, and 31 μm. (**c**) Packaged sensor.

**Figure 3 sensors-23-04841-f003:**
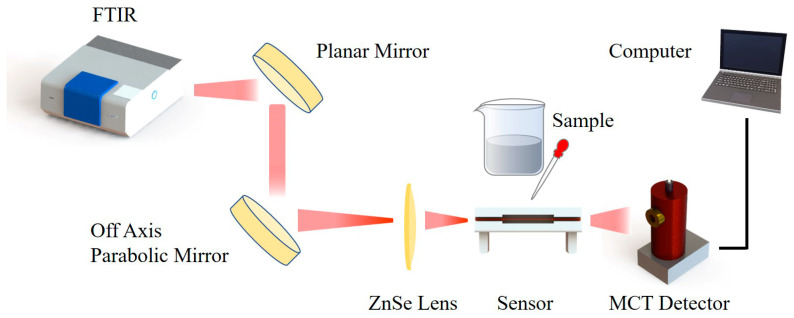
Diagram of liquid sensing platform.

**Figure 4 sensors-23-04841-f004:**
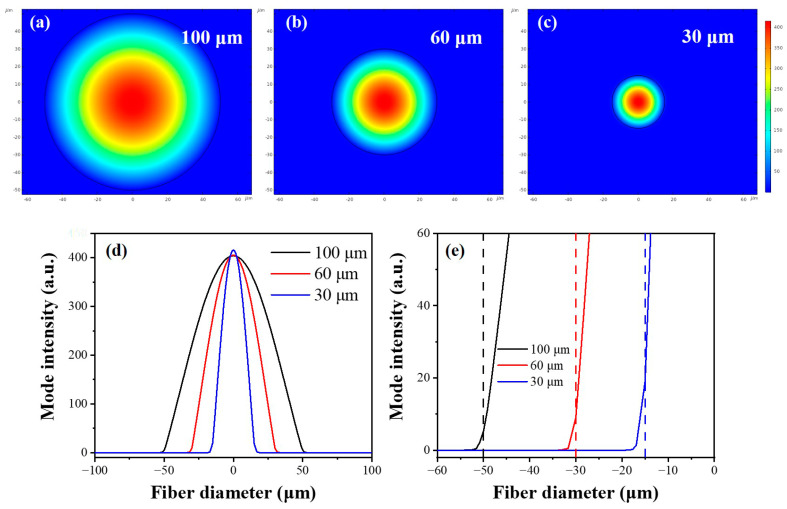
Modes and evanescent wave intensity in fibers with different diameters. Fundamental mode in fiber with a diameter of (**a**)100 μm, (**b**) 60 μm, and (**c**) 30 μm. (**d**) Energy distribution on the axis of the cross-section. (**e**) Evanescent wave intensity at the interface. The dashed lines represent the boundary positions of different fibers located at 25, 30, and 50 μm.

**Figure 5 sensors-23-04841-f005:**
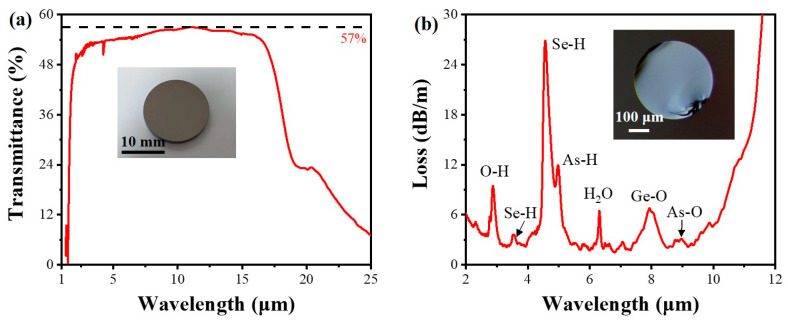
Transmission spectrum and loss spectrum of GAST glass and fiber, respectively. (**a**) Infrared transmission spectrum of GAST glass from 1.5 to 25 μm. Inset shows the polished sample with a thickness of 1.5 mm. (**b**) Loss spectrum of the optical fiber tested by the cut-off method. Inset is the end face of the fiber.

**Figure 6 sensors-23-04841-f006:**
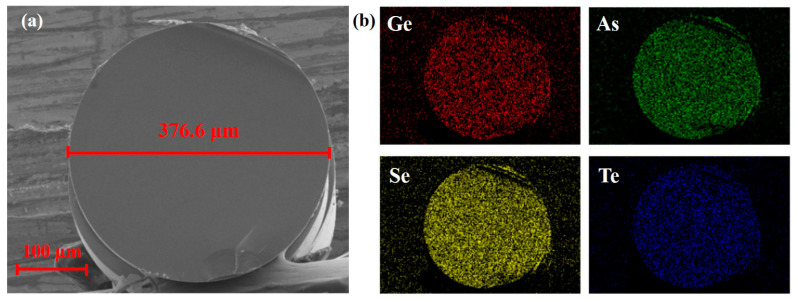
SEM-EDS results of the fiber end face. (**a**) The SEM of the fiber end without the PPSU layer. (**b**) Distribution of Ge, As, Se, and Te in the fiber end.

**Figure 7 sensors-23-04841-f007:**
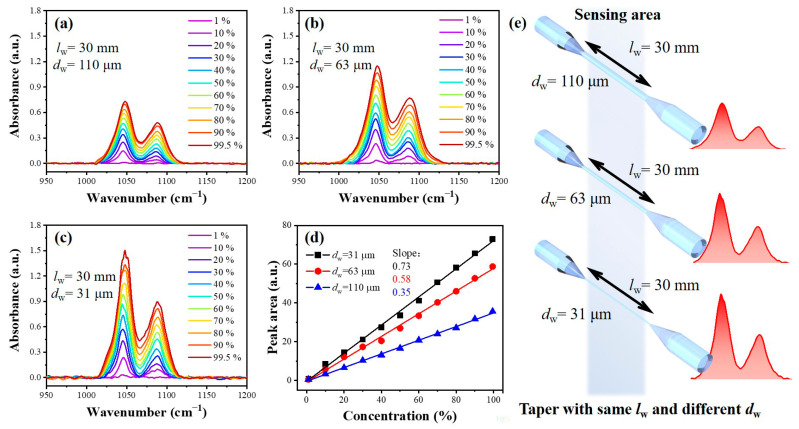
Sensing performance of sensors. Ethanol sensing results of fiber with (**a**) *d*_w_ = 110 μm, (**b**) *d*_w_ = 63 μm, and (**c**) *d*_w_ = 31 μm. (**d**) Sensitivities of different fiber sensors fitted by area of absorption peaks at 1047 and 1088 cm^−1^. (**e**) Schematics of tapered fibers with different *d*_w_. Red peaks represent evanescent wave absorption spectra of 50 vol.% ethanol solution detected by fibers of different *d*_w_.

**Figure 8 sensors-23-04841-f008:**
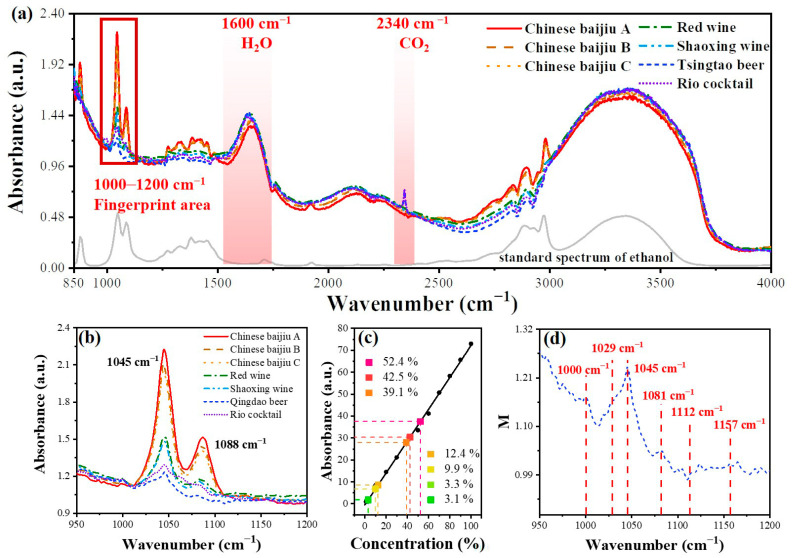
The application for alcohol detection. (**a**) Evanescent wave absorption spectra of seven alcohols, in which ethanol, water, and carbon dioxide can be detected. (**b**) Absorption spectra of alcohols in fingerprint area. (**c**) Ethanol concentrations measured in alcohols using a fitted sensitivity curve. (**d**) Characteristic absorption peaks in the beer, showing the existence of maltose.

**Table 1 sensors-23-04841-t001:** The composition of Ge, As, Se, and Te measured by EDS.

Element	Nominal Composition (mol.%)	Measured Composition (mol.%)	Δ (mol.%)
Ge	10	10.1	0.1
As	30	29.8	0.2
Se	40	39.9	0.1
Te	20	20.2	0.2

**Table 2 sensors-23-04841-t002:** Detected and nominal alcoholicity in alcohols.

Alcohols	Measured (vol.%)	Calibrated (vol.%)	Δ (vol.%)
Chinese baijiu A	52.4	54	1.6
Chinese baijiu B	42.5	42	0.5
Chinese baijiu C	39.1	40	0.8
Red wine	12.4	12.5	0.1
Shaoxing wine	9.9	10.5	0.6
Tsingtao beer	3.3	3.1	0.2
Rio cocktail	3.1	3	0.1

## Data Availability

The raw data required to reproduce these results cannot be shared at this time as the data also form part of an ongoing study.

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
