# Peer review of "Infrared Evanescent Wave Sensing Based on a Ge10As30Se40Te20 Fiber for Alcohol Detection"

_sensors, 2023, doi:10.3390/s23104841_

Round 1

Reviewer 2 Report

The paper presents a novel approach to detecting ethanol using an infrared evanescent wave sensor based on chalcogenide fiber. The experimental part of the paper is well-written and the results are convincing. However, the analytical part of the paper is weak and could be improved.

The authors claim that their sensor is a tapered fiber sensor, but the sensing element is a thin uniform fiber that does not change its diameter. By definition, the taper is a part of the fiber that changes its diameter. The authors should provide a more accurate description of their sensor to avoid confusion.

Furthermore, the authors ignore the influence of changes in the refractive index of the liquid containing ethanol on the mode distributions. Both absorption and refractive sensors are known in the field of optical ethanol sensors, and the authors should address both types of sensors in their analysis.

The authors also claim that alcohol can have health benefits, but there is no scientific evidence to support this claim. The authors should remove this statement or provide appropriate scientific references to support it.

Additionally, the authors refer to a reference [37] that is not appropriate to support their attenuation measurement methodology. They should provide a more relevant reference or describe their methodology with more accuracy.

The authors stated that the temperature used for the tapering was "suitable," but they should provide the exact temperature or temperature range used.

The authors claim that the evanescent wave increases as the fiber diameter decreases, but this statement should be supported by either analysis or reference. Equation (1) relates the penetration depth to the wavelength and not to the diameter of the fiber.

The authors used COMSOL simulations to study the fundamental modes and intensity of the evanescent wave in fibers of different diameters. However, they did not analyze the evanescent field within the area of fiber narrowing, where mode conversion takes place, and the tapering ratio can significantly influence sensor sensitivity. Furthermore, the sensing fiber is multimode, while analysis was performed for only one mode. This omission could be addressed in a future version of the paper.

Finally, the authors selected a spectral range around 1000 1/cm to test ethanol concentrations, but did not explain why this particular spectral fingerprint was selected. The authors should provide a justification for their choice of spectral range.

In summary, this paper presents a novel approach to the detection of ethanol using an infrared evanescent wave sensor based on chalcogenide fiber. The experimental results are convincing, but the analytical part of the paper could be improved. The authors should provide a more accurate description of their sensor, address both absorption and refractive sensors in their analysis, and provide appropriate scientific references to support their claims. They should also provide a more relevant reference or describe their attenuation measurement methodology more accurately, provide the exact temperature or temperature range used for tapering, and support their claim that the evanescent wave increases as the fiber diameter decreases. Additionally, the authors should address the mode conversion that takes place within the area of narrowing of fibers in future studies and provide a justification for their choice of spectral range.

Only minor revision required.

Round 2

Reviewer 2 Report

In the current form, the paper is acceptable for publication, except for Fig. 7e. In Fig. 7e the illustrations suggest that the energy guided by tapers of larger diameter is lower than the energy guided by taper of smaller diameter, which is not true.
